# Factors Affecting Quality of Life of Caregivers of Patients with Heart Failure

**DOI:** 10.3390/healthcare13121363

**Published:** 2025-06-06

**Authors:** Maria Polikandrioti, Athanasia Tsami

**Affiliations:** Department of Nursing, University of West Attica, 122 43 Egaleo, Greece

**Keywords:** quality of life, SF36, heart failure

## Abstract

**Introduction:** The clinical syndrome of heart failure (HF) is progressive and disabling for patients who rely on their caregivers for help and support. The caregiving role is inducing major changes in the personal, social, and family life of caregivers and entails a heavy emotional and physical burden, which in turn negatively affects their quality of life (QoL). **Purpose**: The purpose of this study was to explore the QoL of caregivers of patients with HF as well as the associated caregivers’ and patients’ characteristics. **Material and Methods**: The sample of the study included 340 caregivers along with their hospitalized patients. Data collection was performed by the method of the interview using the questionnaire “SF-36 Health Survey (SF-36)” to assess caregivers’ QoL. **Results:** Τhe physical QoL component of caregivers was found to be statistically significantly associated with the type of relationship with their patient (*p* = 0.001), age (*p* = 0.001), level of education (*p* = 0.001), occupation (*p* = 0.001), information about HF patients (*p* = 0.001), worry about finances (*p* = 0.001), and insecurity about the future (*p* = 0.001). The mental QoL component of caregivers was found to be statistically significantly associated with the type of relationship with their patient (*p* = 0.001), gender (*p* = 0.009), age (*p* = 0.001), level of education (*p* = 0.001), occupation (*p* = 0.001), frequency of visits to hospitalized patient (*p* = 0.001), information about HF (*p* = 0.029), anxiety about patients’ self-care (*p* = 0.001), worry about finances (*p* = 0.001), and insecurity about the future (*p* = 0.001). In terms of HF patient’s characteristics, the physical QoL component of caregivers was found to be statistically significantly associated with the patients’ age (*p* = 0.001), patients’ education level (*p* = 0.001), patients’ occupation (*p* = 0.006), patients’ family history of cardiac disease (*p* = 0.006), and patients’ self-reported symptom management before hospital admission (*p* = 0.022). The mental QoL component of caregivers was found to be statistically significantly associated with the patients’ education level (*p* = 0.020), the patients’ NYHA stage (*p* = 0.001), prior hospitalization (*p* = 0.001), the patients’ family history of cardiac disease (*p* = 0.012), and the patients’ self-reported symptom management before admission (*p* = 0.001). **Conclusions**: In-depth understanding factors affecting QoL in caregivers with HF may enhance plans and actions to attain healthcare goals among societies globally.

## 1. Introduction

Heart failure (HF) consists of a progressive clinical syndrome affecting approximately 64 million individuals globally [1,2]. Informal caregivers play a vital role on addressing HF patients’ physical, emotional, and practical needs [3,4].

An informal caregiver is a family member that provides unpaid care to the loved person suffering from an illness. According to recent estimates, caregiver incidence is expanding at an alarming rate in Europe. For example, in 2021, France reported 9.3 million caregivers, including 4.3 million caregivers of the elderly. In 2017, 20% of individuals above 50 years old were informal caregivers in many European countries [5]. In the USA, 39.8 million informal caregivers provide care in patients with chronic disease, such as HF [6].

In HF, the caregiving role involves complex, excessive, and constant responsibilities for patients’ care [4]. Factors that determine caregivers’ QoL include the amount of time dedicated to care, patients’ comorbidities, disease exacerbation, collaboration with health professionals, the quality of the relationship, communication between the dyad (caregivers, patients), and the patient’s age and functional status [7,8,9]. Furthermore, the majority of HF patients may be able to not afford a paid caregiver and depend on their informal caregivers for help in daily activities or in more practical issues, such as transportation to healthcare services [8].

HF involves frequent visits to healthcare services [1], which is an independent predictor of QoL deterioration among family caregivers [7]. In the USA, HF leads to over than one million hospitalizations annually [1]. It is estimated that approximately 1 in 4 HF patients are readmitted after hospital discharge within 30 days [10]. Hospitalization is a stressful experience for informal caregivers, as they are required to adapt to an unfamiliar environment, spend many hours in the hospital, deliver bedside care to patients, and encounter several challenges related to this clinical syndrome. Though caregivers provide high-intensity care, they usually do not receive any professional care during their hospital stay [11,12]. The World Health Organization demonstrates the need to empower informal care as a fundamental element of the European population [13].

QoL measurement detects signs of deterioration early in the physical and mental health of caregivers or reveals the associated factors. The ultimate goal of QoL measurement is to relieve or prevent caring burden [5].

In an attempt to address these issues, the aim of the present study was to explore QoL in family caregivers of hospitalized patients with HF.

## 2. Material and Methods

### 2.1. Design, Setting, and Period of the Study

In the present study, 340 caregivers were enrolled with their hospitalized HF patients (dyads) in public hospitals in Athens from 2021 to 2024. In this cross-sectional study, participants were selected using the method of convenience sampling.

### 2.2. Inclusion and Exclusion Criteria of the Sample

Criteria for inclusion in the study (patients, caregivers) were as follows: (i) ability to write, read, and comprehend Greek language and (ii) ability to read and sign the informed consent form. For patients, the exclusion criteria were hospitalization for other comorbidities apart from HF.

### 2.3. Data Collection and Procedure

Data were completed by the method of interview. The process of filling out the questionnaire lasted between 20 and 30 min and took place when caregivers had no duties and activities for patient care.

### 2.4. Research Instrument

Data were collected using the 36-Item Short Form Survey (SF-36) scale, including patients’ and caregivers’ characteristics. In terms of caregivers’ characteristics, we recorded the following: relation with patient, gender, age (years), education, occupation, level of information about HF, financial worries, insecurity about future, frequency of visits to hospitalized patients, and anxiety about patients’ self-care. Regarding patients’ characteristics, we recorded the following: gender, age, education, occupation, NYHA classification, prior hospitalization, family history of cardiac disease, and patients’ belief about prevention of symptom deterioration before hospital admission.

#### Measurement of Caregivers’ QoL

For QoL assessment of caregivers, we used the “SF-36 Health Survey (SF-36)” scale, which was created by Ware and colleagues. The SF-36 scale consists of 36 questions that assess the following 8 dimensions: physical functioning, role-physical, bodily pain, general health, energy/fatigue, social functioning, emotional role, and emotional wellbeing. Participants answered each question on Likert-type scales. The summary of scale assessed two components: (a) the physical component summary (PCS) and (b) the mental component summary (MCS). The physical component summary (PCS) includes physical functioning, physical role, bodily pain, and general health, while the mental component summary (MCS) includes energy/fatigue, social functioning, emotional role, and emotional wellbeing.

Higher scores indicate better QoL [14,15,16].

### 2.5. Ethical Considerations

The present study was approved by the research committee of public hospital (Number: 476/3585 on 10 February 2021). Participants were informed by the researcher for the purposes of the study, and their written informed consent was obtained. Data collection guaranteed anonymity and confidentiality. All subjects were informed of their rights to refuse or discontinue participation in the study according to the ethical standards of the Declaration of Helsinki (1989) of the World Medical Association.

### 2.6. Statistical Analysis

Categorical data are presented with absolute and relative frequencies (%), while continuous data are presented with mean, standard deviation, median, and interquartile range. Normality of the data was tested with Kolmogorov–Smirnov criterion and graphically with Q-Qplots and Histograms. Non-parametric tests Mann–Whitney, Kruskal–Wallis test were used to test for the association between caregivers’ and patients’ characteristics and caregivers’ QoL. In addition, multiple linear regression was performed to estimate the effect of caregivers’ and patients’ characteristics on caregivers’ QoL. Results are presented as β regression coefficients and 95% confidence interval (95%CI). The observed level of 5% was considered statistically significant. All statistical analyses were performed with SPSS version 28 (SPSS Inc., Chicago, IL, USA).

## 3. Results

### 3.1. Caregivers Description

Table 1 presents the caregivers’ sample description. A total of 71.8% were women, 68.8% were spouses, 45.6% were over 60 years of age, 38.8% had secondary education, 25.6% were pensioners, and 24.1% were above 70 years old.

Moreover, 39.4% were very informed about the patient’s HF, 31.8% were very worried about finances, 36.5% were feeling very insecure about the future, 46.5% were staying in the hospital during the patient’s hospitalization, and 63.2% reported anxiety about the patient’s self-care.

### 3.2. Patients Description

Table 2 presents the patients’ sample description. Men accounted for the 66.5% of the patients, 85.6% were over 60 years of age, 44.1% had primary education, and 72.1% were retired. Furthermore, NYHA stage III had 47.1% of the sample and stage IV 20.3%. Previous hospitalization had 75.6% of the sample and family history of cardiac disease 72.1%. Lastly, only 45.0% stated they managed symptoms before hospital admission.

### 3.3. Caregiver’s QoL

Table 3 presents the results regarding the caregivers’ quality of life. The summary of physical and mental components indicates a worse QoL, with medians 33.2 and 46.2, respectively, in a range of score 0–100.

### 3.4. Characteristics of Caregivers Associated with Caregivers’ QoL

Table 4 presents the associations between caregivers’ characteristics with their QoL. The QoL physical component summary of caregivers was found to be statistically significantly associated with the type of relationship with their patient (*p* = 0.001), age (*p* = 0.001), the level of education (*p* = 0.001), occupation (*p* = 0.001), the level of information about HF of patients (*p* = 0.001), how much they worried about finances (*p* = 0.001), and how insecure they felt about the future (*p* = 0.001). Specifically, caregivers who were spouses of patients had worse QoL in the physical component (median 31.6) compared to caregivers who were children of patients (median 38.2). Caregivers above 70 years old had worse QoL in the physical component (median 26.9) compared to caregivers ≤40 years old (median 38.9), 41–50 (median 37.9), 51–60 (median 36.6), and 61–70 (median 30.1). Caregivers with primary education had worse QoL in the physical component (median 27.1) compared to caregivers with higher education. In addition, there was a worse QoL in the physical component in caregivers who were retired (median 28.9), those who were little informed about HF of patient (median 29.5), those who worried a lot about finances (median 29.6), and those who felt very insecure about the future (median 29.5).

The QoL mental component summary of caregivers was found to be statistically significantly associated with the type of relationship with the patient (*p* = 0.001), gender (*p* = 0.009), age (*p* = 0.001), education level (*p* = 0.001), occupation (*p* = 0.001), the frequency of visits to the hospitalized patient (*p* = 0.001), the level of information about HF of patients, whether they reported anxiety about the patient’s care (*p* = 0.001), how much they worried about finances (*p* = 0.001), and how insecure they felt about the future (*p* = 0.001). Specifically, caregivers who were spouses of patients had a worse QoL in the mental component (median 43.1) compared to caregivers who were children of patients (median 49.6). Female caregivers had a worse QoL in the mental component (median 44.9) compared to males (median 48.1). There was a worse QoL in mental component in caregivers above 70 years (median 41.0) compared to caregivers ≤40 years old (median 50,3), 41–50 (median 49.9), 51–60 (median 50.1), and 61–70 (median 42.2). Caregivers with primary education had a worse QoL in the mental component (median 28.7) compared to caregivers with higher levels of education. In addition, there was a worse QoL in mental component in caregivers who were retired (median 38.8), those who stayed in the hospital with the patient (median 42.1), those who reported anxiety about the patients’ self-care (median 42.4), those who were little informed about HF of patients (median 42.8), those who were very worried about finances (median 37.1), and those who felt very insecure about the future (median 37.5).

### 3.5. HF Patient’s Characteristics Associated with Caregivers’ QoL

Table 5 presents the associations between patients’ characteristics and caregivers’ QoL. The QoL physical component summary of caregivers was found to be statistically significantly associated with the patients’ age (*p* = 0.001), patients’ education level (*p* = 0.001), patients’ occupation (*p* = 0.006), patients’ family history of cardiac disease (*p* = 0.006), and patients’ report about their management of symptoms before hospital admission (*p* = 0.022). Specifically, caregivers of patients over 70 years of age (median 31.4), caregivers of patients with primary education (median 31), caregivers of retired patients (median 31.2), caregivers of patients with a family history of cardiac disease (median 32.1), and caregivers of patients who could not manage symptoms before hospital admission (median 32.5) had a worse QoL in the physical component.

The QoL mental component summary of caregivers was found to be statistically significantly associated with the patients’ education level (*p* = 0.020), patients’ NYHA stage (*p* = 0.001), patients’ prior hospitalization, patients’ family history of cardiac disease (*p* = 0.001 and *p* = 0.012), and patients’ report about their management of symptoms before hospital admission (*p* = 0.001). Specifically, caregivers of patients with primary education (median 42.7), caregivers of patients with NYHA IV (median 36.6), caregivers of patients with prior hospitalization and a family history of cardiac disease (median 44.2 and 44.5, respectively), and caregivers of patients who could not manage symptoms before hospital admission (median 42) had a worse QoL in the mental component.

### 3.6. Effect of Caregivers’ Characteristics on Caregiver’s Quality of Life

Multiple linear regression in Table 6 was performed to estimate the effect of the caregivers’ and patients’ characteristics on the caregivers’ QoL (independent factors).

Caregivers aged 51–60 years had a 2.70-point-worse physical score than caregivers aged <40 (b = −2.70 95%CI: −5.37–−0.03, *p* = 0.047). Employees caregivers had a 3.19-point-better physical score than unemployed caregivers (b = 3.19 95%CI: 1.28–5.09, *p* = 0.001). Caregivers who were a little informed about patient’s HF had a 2.39-point-worse physical score than caregivers who were very informed (b = −2.39 95%CI: −4.43–−0.35, *p* = 0.022). Caregivers who were taking care of patients aged 61–70 years had a 3.47-point-worse physical score than caregivers who were taking care of patients aged <60 years (b = −3.47 95%CI: −5.75–−1.20, *p* = 0.003). Caregivers who were taking care of patients with no family history of cardiac disease had a 1.67-point-better physical score than caregivers who were taking care of patients with a family history (b = 1.67 95%CI: 0.17–3.17, *p* = 0.030).

Furthermore, caregivers with university education had a 4.6-point-better mental score than caregivers with primary education (b = 4.60 95%CI:1.06–8.13, *p* = 0.011). Pensioners caregivers had a 4.95-point-worse mental score than unemployed caregivers (b = −4.95 95%CI: −8.41–−1.48, *p* = 0.005). Caregivers who were feeling a little insecure about the future had a 5.63-point-better mental score than caregivers who were feeling very insecure (b = 5.63 95%CI: 1.90–9.35, *p* = 0.003). Caregivers who were taking care of patients with no prior hospitalization had a 2.78-point-better mental score than caregivers who were taking care of patients who were previously hospitalized (b = 2.78 95%CI: 0.17–5.38, *p* = 0.037).

## 4. Discussion

The results of the present study showed that both the mental and physical component of QoL were impacted. The physical component was most impacted, with a mean score of 33.1 ± 6.9. In the SF-36 scale, higher scores indicate better health status, and the mean score of 50 is articulated as a normative value for scales.

In terms of demographic characteristics, 71.8% were women, 68.8% were spouses, 45.6% were over 60 years of age, 38.8% had secondary education, 25.6% were pensioners, and 24.1% were above 70 years old. According to typical profile of family caregivers, they tend to be mostly women, to have an average age of 53 to 56, to be housekeepers, to have low education levels, and to not work outside the home, and in most cases, they are daughters or spouses of the person under their care [7,17]. Accordingly, a relevant study in Korea showed that two-thirds of caregivers were women, and more than 50% had no job [4]. In Europe countries, approximately 61% of informal caregivers are women within the family [13]. Similarly, in Greece, three out of four informal caregivers are females [18].

Regarding demographic characteristics, there was a diminished QoL in both the physical and mental components in caregivers of primary education and the retired ones, while female caregivers had worse QoL only in the mental component. Rico-Blázquez et al. [13] support that caregiver’s age, gender, and marital status are the determinants of burden on their health, while Kim et al. [19] state a correlation between education, economic factors, and overall QoL. The low socioeconomic level is an indicator of diminished ability to pay for treatment, to effectively communicate with the health system, and to comprehend the recommended therapy [6]. Moreover, there are noticeable gender inequalities among female caregivers in chronic illness in experiences of anxiety and depression, sleep disturbances, increased pain, increased discomfort, and diminished health state. In Mediterranean countries, men serve as primary family caregivers only when women’s caregiving resources are exhausted. During hospitalization, caregivers act daily under stress in a demanding environment [19] with complex patient issues [19] and experience discomfort due to an absence of a space used for meetings or for some rest [20]. There was a better QoL in both components in caregivers under 60 years old. A relevant study showed that caregivers’ QoL was decreased when their age was increased [4].

In the present study, 45.9% of participants stayed in the hospital, which is attributed to the strong traditional family bonds or to cultural influences on care provision patterns. Likewise, in Northern Greece, the majority of caregivers spent over 17 h/day at the hospital [18]. Dysfunctions of healthcare facilities, nursing staff shortages, and the fear of caregivers that being away from the hospitalized person may result in unfulfilled needs explain their reluctance to leave the hospital [18,21]. Those staying in the hospital had a worse QoL in the mental component.

Family caregivers little informed about HF reported a diminished QoL in both the physical and mental components. In some cases, caregivers may not recall or deeply understand the information, especially when provided in busy hospital circumstances. Inadequate information or information leading to false hope triggers anxiety [20,22]. Consequently, these factors partially explain their low QoL.

Also, there was a diminished QoL both in the physical and mental components in caregivers who declared financial worries and insecurity about future. Unfortunately, in Greece, there are no public long-term care facilities to support patients with chronic illness who in turn are admitted to hospitals, thus imposing an extra burden on caregivers. Additionally, caregivers fail to receive financial support from institutional services due to an absence of relevant legislation [23,24]. Uncertainty is associated with adverse emotional outcomes (fear, doubts, misunderstandings), psychological outcomes (depression, anxiety, negative coping mechanisms), and financial outcomes (long-term financial burden). The nature of a patient’s illness, disease severity, complex medical treatment, and poor prognosis are only some of the reasons that generate uncertainty among caregivers [25].

A noticeable finding is the worse QoL in mental component in caregivers who declared anxiety about the patients’ self-care. One possible explanation might be that caregivers perceive the burden of patients ‘dependency on them [13]. Elderly patients with chronic diseases are more frequently hospitalized either for diagnostic evaluation or due to illness consequences. As a general rule, the length of hospital stay is shortening, and patients may not achieve a secure health status before hospital discharge. Therefore, care becomes more complex, often entailing greater demands on family caregivers [26]. At the transition phase (from hospital to home), a possible contributor to caregivers’ QoL may be insufficient knowledge, skills, and competences about the provision of multifaceted and long self-care management, including symptom monitoring and medication adherence as well as a lack of support and low self-efficacy [2,27]. Notably, in discharge instructions, the role of caregivers is strongly discussed along with strategies to enhance medication adherence, lifestyle modifications, relationships with health professionals, and telephone calls early after hospital leave [27].

In terms of patients’ characteristics, there was a worse QoL in both the physical and mental components in caregivers whose patients were of primary education, had a family history of cardiac disease, and declared no symptom management before hospital admission. Only in the physical component, there was a worse QoL in caregivers whose patients were 71–80 years old and pensioners, while only in the mental component, there was a worse QoL in the caregivers of patients with prior hospitalizations and NYHA stage IV. Patients’ comorbidity, poorer cardiac functioning, and female gender are positively associated with the number of hours per day and the length of time the caregiver provides care [7]. Patients’ symptoms (dyspnea, fatigue), limited dependency, high disability (NYHA IV), polypharmacy, and the unpredictable nature of HF, including exacerbations and frequent rehospitalizations, may cause distress and helplessness in caregivers, thus negatively affecting their QoL [1,28].

The common factor among patients and caregivers that affected caregivers’ QoL both in the physical and mental components was the educational level of each member of this dyad. More in detail, caregivers whose patients had primary education had a worse QoL, and also, caregivers who were of primary education had a worse QoL. Individuals of low educational level are frequently of advanced age, of low socioeconomic status, with more comorbidities, and with higher mortality risk [29]. Multiple regression showed that caregivers with university education had a 4.6-point-better mental QoL score, which is a finding of clinical significance. Educated individuals cultivate skills and attitudes that enable them to better comprehend their needs and make informed decisions. Education per se is a protective determinant of mental health. A year of education leads to a lower likelihood of depression and anxiety through several pathways, such as improved health behavior and knowledge [30].

Improving caregivers’ QoL is important during hospitalization since they are a source of information about patients’ needs, symptoms, and medication adherence as well as the link of communication with the healthcare team [18]. Each finding regarding the caregivers’ mental state illustrates the need for further medical monitoring in order to determine a deterioration or the need for medication [7].

It would be interesting to measure caregivers’ QoL during several phases in illness trajectory, such as diagnosis, when they cope with long term HF management at the transition phase from hospital to home and at the end stage [1]. To the best of our knowledge, data on factors affecting the QoL of caregivers’ with terminally ill HF patients are diminished. Implanted cardiac devices, such as left ventricular assist device (LVAD), which are used in more than one-third of HF patients eligible for heart transplantation, have changed the already known concept of life’s end. However, caregivers of recipients experience burden (mental, physical) or family, social, and economic constraints. Moreover, end-of-life situations, including the deactivation of an implanted cardiac device (defibrillator, pacemaker), impact caregivers’ QoL [31]. Caregivers and terminally ill HF patients need adequate opportunities to discuss prognosis, sudden cardiac death, and other life priorities [32].

Given that HF is a life-threatening disease, the integration of palliative care services is essential to improve patients’ and caregivers’ QoL. The major impact of early palliative care is not only to improve patient’s symptom control and comfort but also to prepare caregivers for the patient’s health deterioration and death prospect. The readiness of caregivers has a positive impact on QoL. Palliative care is recommended by the cardiology associations and may be incorporated at any time in HF care as complementary to traditional management [33]. It is of crucial importance to dispel caregivers’ misconceptions that delay access to this care [34]. More strikingly, technology might increase the effectiveness of informal care provision and enhance QoL on the condition that all caregivers have access to it [1]. Last but not least, it is important to note that the changing landscape in HF is artificial intelligence (AI), which is anticipated to improve the dyad QoL. The algorithms of AI promise HF care transformation by shedding light on early detection and on subclinical deterioration, thus leading to improved, timely clinical decision-making and reduced healthcare expenditures [35].

### Limitations of the Study

The present study was of a cross-sectional design, and there was no evidence of a causal relationship between caregivers’ and patients’ characteristics and QoL. The method of convenience sampling is not representative of all HF caregivers living in Greece, thus limiting the generalizability of the results. More in detail, convenience sampling involves collecting samples from easily accessible individuals and not through a randomized process, where every individual has an equal chance of inclusion in the sample. Convenience sampling may not reflect the characteristics of the entire caregiver population. Furthermore, there was no second measurements at hospital discharge.

The SF-36 scale is a widely used tool that will help to compare the results with other research studies on a global scale.

## 5. Conclusions

There was a worse QoL in both the physical and mental components in caregivers who were spouses of patients, were aged above 70 years, had primary education, were retired, were little informed about HF, and those who worried a lot about their finances and had insecurity about the future. There was a worse QoL only in the mental component in caregivers who stayed in the hospital with the patient, those who reported anxiety about patients’ self-care, and female caregivers.

Patients’ characteristics associated with worse caregivers’ QoL in both the physical and mental components were primary education, a family history of cardiac disease, and their self-report that they did not manage symptoms before hospital admission. There was a worse QoL only in the physical component in caregivers whose patients aged over 70 years of age and the retired ones. There was a worse QoL only in mental component in caregivers whose patients were of NYHA IV and had a prior hospitalization.

### Future Perspectives

From a clinical perspective, it is important for healthcare professionals to acknowledge the significance of QoL assessment. A first measurement is essential at HF patients’ hospital admission in order identify caregivers with an already established QoL deterioration. According to the results of the first measurement, healthcare professionals provide support to promote caregivers’ health. At hospital discharge, a second measurement is necessary to reassess QoL levels and schedule early monitoring for caregivers with a diminished QoL.

From a healthcare policy perspective, intervention programs that focus on caregivers’ QoL improvement are essential. More in detail, prescheduled actions such as support for caregivers’ adaptation to illness trajectory, the provision of alternative resources, psychological counseling, and education about HF patients’ self-care are needed. Future sustainability of informal caregiving resources demands public policy measures to promote their health and to provide financial assistance or a leave of absence from work.

From a research perspective, caregivers belong to unrepresented groups, and available data regarding factors associated with the person that provides care as well as tools on caregivers’ QoL with HF patients are limited.

“*Care to caregivers*” is obviously one of the most effective pathways to prevent their QoL deterioration and reduce costly rehospitalizations of HF patients.

## Figures and Tables

**Table 1 healthcare-13-01363-t001:** Sample description of caregivers (n = 340).

	n (%)
Relation to patient	
Child	106 (31.2%)
Spouse	234 (68.8%)
Gender	
Male	96 (28.2%)
Female	244 (71.8%)
Age (years)	
<40	48 (14.1%)
41–50	75 (22.1%)
51–60	62 (18.2%)
61–70	73 (21.5%)
>70	84 (24.1%)
Education	
Primary	80 (23.5%)
Secondary	132 (38.8%)
University	128 (37.7%)
Occupation	
Employed	167 (49.1%)
Household	86 (25.3%)
Pensioner	87 (25.6%)
Informed about patient’s HF	
Very	134 (39.4%)
Enough	145 (42.6%)
A little	61 (17.9%)
Experience of financial worries	
Very	108 (31.8%)
Enough	154 (45.3%)
A little	52 (15.3%)
Not at all	26 (7.6%)
Insecurity about future	
Very	124 (36.5%)
Enough	153 (45.0%)
A little	56 (16.5%)
Not at all	7 (2.1%)
Frequency of visits	
Stay in hospital	158 (46.5%)
Once a day	121 (35.6%)
Twice a day	18 (5.3%)
Day by day	43 (12.6%)
Anxiety about patients’ self-care (yes)	215 (63.2%)

**Table 2 healthcare-13-01363-t002:** Sample description of patients (n = 340).

	n (%)
Gender	
Male	226 (66.5%)
Female	114 (33.5%)
Age (years)	
41–50	9 (2.7%)
51–60	40 (11.8%)
61–70	93 (27.4%)
71–80	150 (44.1%)
>80	48 (14.1%)
Education	
Primary	150 (44.1%)
Secondary	114 (33.5%)
University	76 (22.4%)
Prior hospitalization (yes)	257 (75.6%)
Family History of Cardiac Disease (yes)	245 (72.1%)
Manage symptoms before admission (yes)	153 (45.0%)
NYHA	
I	14 (4.1%)
II	97 (28.5%)
III	160 (47.1%)
IV	69 (20.3%)
Occupation	
Employed	59 (17.4)
Household	36 (10.5%)
Pensioner	245 (72.1%)

**Table 3 healthcare-13-01363-t003:** Caregiver’s QoL (n = 340).

SF36 Quality of Caregiver’s Life	Mean (SD)	Median (IQR)
Physical Component Summary (PCS)	33.1 (6.9)	33.2 (27.5–39.0)
Mental Component Summary (MCS)	45.5 (12.5)	46.2 (35.4–54.7)

SD: Standard deviation, IQR: interquartile range.

**Table 4 healthcare-13-01363-t004:** Caregiver’s characteristics associated with their QoL.

	Caregiver Physical Component (PCS)			Caregiver Mental Component (MCS)		
Caregiver’s Characteristics	Mean	Median	*p*-Value	Mean	Median	*p*-Value
	(SD)	(IQR)		(SD)	(IQR)	
Relation with patient			0.001			0.001
Child	36.1 (6.0)	38.2 (31.5–40.5)		49.3 (12.6)	49.6 (40.4–60.5)	
Husband/Wife	31.8 (6.9)	31.6 (26.5–37.2)		43.9 (12.1)	43.1 (34.3–52.1)	
Gender			0.078			0.009
Male	34.1 (6.8)	35.1 (28.8–39.9)		48.7 (11.3)	48.1 (38.7–58.4)	
Female	32.7 (6.9)	32.6 (27.3–38.7)		44.3 (12.7)	44.9 (34.4–53.2)	
Age (years)			0.001			0.001
≤40	37.1 (5.9)	38.9 (32.4–41.0)		48.9 (13.3)	50.3 (38.1–61.0)	
41–50	36.5 (5.6)	37.9 (32.4–40.7)		48.7 (12.3)	49.9 (39.8–59.3)	
51–60	34.8 (5.7)	36.6 (30.6–39.0)		49.6 (12.6)	50.1 (42.7–60.9)	
61–70	31.2 (6.1)	30.1 (26.5–34.9)		41.7 (11.7)	42.2 (33.9–48.8)	
>70	28.4 (6.6)	26.9 (23.7–33.6)		41.2 (10.6)	41.0 (33.4–49.8)	
Education			0.001			0.001
Primary	28.4 (5.7)	27.1 (24.4–31.5)		39.4 (11.3)	38.7 (31.3–46.8)	
Secondary	32.6 (6.8)	32.9 (27.4–37.7)		44.3 (11.3)	45.9 (34.7–52.0)	
University	36.4 (5.8)	38.1 (32.3–40.5)		50.5 (12.5)	51.0 (40.9–61.0)	
Occupation			0.001			0.001
Household	30.2 (5.7)	30.1 (26.1–33.8)		42.8 (10.9)	42.3 (34.6–49.1)	
Employee	36.7 (5.8)	38.4 (32.6–40.6)		49.9 (12.2)	50.8 (40.8–60.5)	
Pensioner	29.4 (6.5)	28.9 (24.9–34.5)		40.2 (11.7)	38.8 (31.3–48.1)	
Frequency of visits			0.278			0.001
Stay in hospital	32.5 (7.2)	32.3 (26.8–37.7)		43.1 (12.5)	42.1 (34.2–51.2)	
Once/Twice a day	33.5 (6.6)	33.4 (27.9–39.0)		46.8 (12.0)	48.3 (37.9–55.8)	
Day by day	33.9 (6.8)	37.5 (27.9–39.6)		50.6 (12.2)	49.0 (42.7–63.2)	
Informed about patient’s HF			0.001			0.029
Very	35.2 (6.4)	36.6 (31.1–40.3)		47.6 (13.2)	49.4 (38.2–58.4)	
Enough	32.2 (6.8)	31.8 (26.6–38.2)		44.3 (12.2)	44.2 (34.8–53.1)	
A little	30.4 (6.8)	29.5 (24.7–36.4)		43.6 (10.7)	42.8 (34.4–51.6)	
Anxiety about patients’ selfcare			0.267			0.001
Yes	32.8 (7.0)	33.0 (26.9–38.8)		43.5 (12.2)	44.4 (34.2–51.7)	
No	33.7 (6.7)	34.3 (29.3–39.4)		49.1 (12.2)	49.7 (38.4–60.3)	
Financial worries			0.001			0.001
Very	30.1 (6.7)	29.6 (25.1–34.0)		38.0 (11.0)	37.1 (28.8–45.9)	
Enough	33.8 (6.8)	33.8 (29.2–39.4)		46.9 (11.0)	47.4 (38.4–54.2)	
A little/Not at all	35.9 (5.8)	38.0 (31.7–40.2)		53.0 (11.8)	53.2 (47.1–63.6)	
Insecurity about future			0.001			0.001
Very	30.0 (7.1)	29.5 (25.0–33.8)		38.9 (11.6)	37.5 (30.0–47.0)	
Enough	34.2 (6.3)	34.2 (29.5–39.1)		47.2 (11.3)	47.4 (38.4–55.8)	
A little/Not at all	36.6 (5.3)	38.2 (34.9–40.3)		54.4 (10.1)	54.7 (48.4–63.1)	

**Table 5 healthcare-13-01363-t005:** Patient’s characteristics associated with their caregivers’ QoL.

	Caregiver Physical Component (PCS)			Caregiver Mental Component (MCS)		
Patient’s Characteristics	Mean (SD)	Median (IQR)	*p*-Value	Mean (SD)	Median (IQR)	*p*-Value
Gender			0.705			0.996
Male	33.0 (7.0)	33.4 (27.5–39.0)		45.5 (13.3)	45.8 (34.4–58.1)	
Female	33.3 (6.8)	33.0 (28.4–39.2)		45.5 (10.7)	46.3 (36.9–52.1)	
Age			0.001			0.263
≤60	37.1 (5.9)	38.7 (32.5–41.1)		48.7 (12.4)	50.3 (36.7–59.1)	
61–70	33.3 (6.4)	33.0 (28.4–38.9)		45.7 (12.9)	46.7 (35.6–53.1)	
71–80	31.8 (6.8)	31.4 (26.5–37.3)		44.4 (12.0)	44.8 (34.9–52.8)	
>80	32.6 (7.5)	32.7 (25.4–39.0)		45.6 (13.2)	45.2 (35.4–55.8)	
Education			0.001			0.020
Primary	31.5 (6.5)	31.0 (26.2–37.6)		43.4 (12.0)	42.7 (35.1–51.5)	
Secondary	33.9 (6.7)	34.2 (29.3–39.4)		47.7 (12.7)	47.9 (37.5–59.0)	
University	35.0 (7.2)	36.9 (29.6–40.3)		46.4 (12.6)	48.9 (36.3–54.2)	
Occupation			0.006			0.086
Household	33.2 (6.6)	34.3 (26.8–39.2)		47.7 (12.3)	50.7 (37.1–55.9)	
Employee	35.9 (6.5)	37.1 (30.9–40.5)		47.9 (14.1)	49.8 (35.7–59.8)	
Pensioner	32.5 (6.9)	32.6 (26.7–38.2)		44.7 (12.1)	44.9 (35.2–53.0)	
NYHA			0.944			0.001
I-II	33.1 (6.3)	33.2 (28.9–37.7)		47.7 (11.4)	48.4 (39.4–54.7)	
III	33.1 (7.4)	33.7 (26.6–39.5)		46.9 (12.1)	47.2 (36.7–57.4)	
IV	33.3 (6.7)	32.2 (28.2–39.0)		38.7 (13.0)	36.6 (28.1–47.8)	
Prior hospitalization			0.137			0.001
Yes	32.8 (7.0)	32.6 (26.7–38.9)		44.0 (12.3)	44.2 (34.9–52.1)	
No	34.1 (6.5)	35.4 (29.3–39.3)		50.0 (12.0)	51.3 (41.6–59.9)	
Family Cardiac History			0.006			0.012
Yes	32.5 (6.8)	32.1 (26.8–38.7)		44.5 (12.2)	44.5 (34.7–52.5)	
No	34.8 (6.9)	36.1 (30.1–39.6)		48.0 (12.9)	49.4 (41.6–58.0)	
Manage symptoms before admission			0.022			0.001
Yes	34.1 (6.6)	34.5 (29.3–39.2)		49.3 (11.6)	49.2 (41.6–59.8)	
No	32.3 (7.0)	32.5 (26.3–38.3)		42.5 (12.4)	42.0 (33.4–51.9)	

**Table 6 healthcare-13-01363-t006:** Impact of caregiver’s and patient’s characteristics on caregiver’s QoL.

	Caregiver Physical		Caregiver Mental	
	Component (PCS)		Component (MCS)	
	b Coef (95%CI)	*p*-Value	b Coef (95%CI)	*p*-Value
Caregiver (Spouse vs. child)	0.32 (−1.89–2.53)	0.774	0.45 (−2.95–3.85)	0.794
Caregiver Gender (female vs. male)	-	-	−0.03 (−2.76–2.70)	0.983
Caregiver Age				
≤40	ref		ref	
41–50	−0.50 (−2.92–1.93)	0.688	1.51 (−2.17–5.19)	0.419
51–60	−2.70 (−5.37–−0.03)	0.047	2.08 (−2.01–6.17)	0.317
61–70	−2.13 (−5.07–0.81)	0.155	−0.61 (−5.21–3.99)	0.796
71–80	−3.69 (−7.27–−0.12)	0.043	3.12 (−2.28–8.51)	0.256
Caregiver Education				
Primary	ref		ref	
Secondary	0.07 (−1.83–1.97)	0.945	2.86 (−0.13–5.85)	0.061
University	0.95 (−1.30–3.19)	0.407	4.60 (1.06–8.13)	0.011
Caregiver Occupation				
Unemployed/Household	ref		ref	
Employee	3.19 (1.28–5.09)	0.001	−0.96 (−4.07–2.14)	0.542
Pensioner	−0.02 (−2.14–2.11)	0.989	−4.95 (−8.41–−1.48)	0.005
Caregiver Frequency of visits				
Stay in hospital	-	-	ref	
Once/Twice a day	-	-	−1.92 (−4.29–0.45)	0.113
Day by day	-	-	−1.12 (−4.92–2.69)	0.563
Caregiver Informed about HF				
Very	ref		ref	
Enough	−1.57 (−3.08–0.05)	0.053	−0.40 (−2.89–2.08)	0.749
A little	−2.39 (−4.43–−0.35)	0.022	2.06 (−1.37–5.49)	0.238
Anxiety about patients’ self care (no vs. yes)	-	-	1.82 (−0.50–4.14)	0.124
Caregiver Insecurity about future				
Very	ref		ref	
Enough	1.38 (−0.20–2.95)	0.086	3.09 (0.58–5.60)	0.016
A little	1.88 (−0.43–4.20)	0.11	5.63 (1.90–9.35)	0.003
Patient Age				
≤60	ref		-	-
61–70	−3.47 (−5.75–−1.20)	0.003	-	-
71–80	−2.54 (−5.20–0.11)	0.06	-	-
>80	−2.66 (−5.89–0.57)	0.107	-	-
Patient Education				
Primary	ref		ref	
Secondary	0.25 (−1.40–1.91)	0.763	−1.60 (−5.87–2.67)	0.46
University	0.65 (−1.22–2.52)	0.495	−0.13 (−3.62–3.36)	0.941
Patient Occupation				
Unemployed/Household	ref		-	-
Employee	1.81 (−0.88–4.50)	0.187	-	-
Pensioner	0.87 (−1.43–3.16)	0.457	-	-
Patient NYHA	-	-		
I–II	-	-	ref	
III	-	-	1.76 (−1.12–4.63)	0.231
IV	-	-	−0.86 (−4.84–3.11)	0.669
Patient prior hospitalization (no vs. yes)	-	-	2.78 (0.17–5.38)	0.037
Patient Family Cardiac History (no vs. yes)	1.67 (0.17–3.17)	0.03	0.57 (−1.75–2.89)	0.627
Patient manage symptoms before admission (no vs. yes)	1.31 (−0.27–2.89)	0.105	−2.47 (−4.98–0.04)	0.054

b coef: b regression coefficient, CI: confidence interval, ref: reference category.

## Data Availability

The original contributions presented in this study are included in the article. Further inquiries can be directed to the corresponding author.

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
