# Peer review of "Factors Affecting Quality of Life of Caregivers of Patients with Heart Failure"

_healthcare, 2025, doi:10.3390/healthcare13121363_

Round 1

Reviewer 1 Report

Comments and Suggestions for Authors

I have carefully read the article titled "Factors affecting quality of life of caregivers of patients with heart failure" that was sent to me for evaluation. My comments, criticisms and suggestions are listed below:
1. It would be good to comment on the impact of early integration of interdisciplinary palliative care services in heart failure patients on the QoL of caregivers.
2. Are there any data on factors affecting QoL in caregivers during the critical period of transition from hospital to home after hospitalization for acute decompensated HF, which is particularly common in long-term HF management?
3. In today's conditions, especially for patients with end-stage HF, when making decisions about advanced treatment modalities such as long-term IV medications, ICD implantation, temporary ventricular support system, implanted pulmonary artery pressure monitoring system, or LVAD implantation, factors affecting the participation of caregivers in the process should be discussed.
4. When a patient with end-stage heart failure reaches the end of life, do they have any data on the factors that determine the caregiver's impact on the process?
5. Discussing the role of remote monitoring, the use of smartphones, and artificial intelligence applications to facilitate the work of caregivers and the health system in the follow-up of heart failure patients will significantly increase the contribution of the article to the literature.
6. It would be appropriate to discuss what are the effects of specific conditions such as immunosuppression, rejection, infection, and biopsy procedures requiring frequent hospitalization on the QoL of caregivers in heart transplant patients who have a unique treatment and care process.
Best Regards

Author Response

Reviewer 1

Thank you very much for your time ! I did all you asked me. I write here all modifications which are better presented within the text in highlight.

  1. It would be good to comment on the impact of early integration of interdisciplinary palliative care services in heart failure patients on the QoL of caregivers.

Response: Given that HF is life threatening disease, the integration of palliative care services is essential to improve patients’ and caregivers’ QoL. The major impact of early palliative care is not only to improve patient’s symptom control and comfort but also to prepare caregivers for patient’s health deterioration and death prospect. Readiness of caregivers has a positive impact on QoL. Palliative care is recommended by the cardiology associations and may be incorporated at any time in HF care as complementary to traditional management [33]. It is of crucial importance to dispel caregivers’ misconceptions that delay access to this care [34]. lines 333-339

  1. Are there any data on factors affecting QoL in caregivers during the critical period of transition from hospital to home after hospitalization for acute decompensated HF, which is particularly common in long-term HF management?

Response: At transition phase, (from hospital to home) a possible contributor to caregiver’ QoL may be insufficient knowledge, skills and competences about provision of multifaceted and long self-care management, including symptom monitoring and medication adherence as well as lack of support and low self-efficacy [2,30]. Notably, in discharge instructions the role of caregivers is strongly discussed along with strategies to enhance medication adherence, lifestyle modifications, relationships with health professionals and telephone calls early after hospital leave [27]. lines 286-293

  1. In today's conditions, especially for patients with end-stage HF, when making decisions about advanced treatment modalities such as long-term IV medications, ICD implantation, temporary ventricular support system, implanted pulmonary artery pressure monitoring system, or LVAD implantation, factors affecting the participation of caregivers in the process should be discussed.
  2. When a patient with end-stage heart failure reaches the end of life, do they have any data on the factors that determine the caregiver's impact on the process?

Response: To the best of our knowledge, data on factors affecting QoL of caregivers’ with terminally ill HF patients are diminished. Implanted cardiac devices such as left ventricular assist device (LVAD) which are used in more than one‐third of HF patients eligible for heart transplantation, have changed the already known concept of life’s end. However, caregivers of recipients experience burden (mental, physical) or family, social and economic constraints. Moreover, end-of-life situations, including deactivation of an implanted cardiac device (defibrillator, pacemaker) impact caregivers’ QoL [31]. Caregivers and terminally ill HF patients need adequate opportunities to discuss prognosis, sudden cardiac death, and other life priorities [32]. lines 324-332

  1. Discussing the role of remote monitoring, the use of smartphones, and artificial intelligence applications to facilitate the work of caregivers and the health system in the follow-up of heart failure patients will significantly increase the contribution of the article to the literature.

Response: Strikingly more, technology might increase effectiveness of informal care provision and enhance QoL on the condition that all caregivers have access to it [1]. Last but not least, it is important to note that the changing landscape in HF is artificial intelligence (AI) which is anticipated to improve the dyad QoL. The algorithms of AI promise HF care transformation by shedding light on early detection and on subclinical deterioration, thus leading to improved, timely clinical decision making and reduced health care expenditures [35]. lines 340-346

  1. It would be appropriate to discuss what are the effects of specific conditions such as immunosuppression, rejection, infection, and biopsy procedures requiring frequent hospitalization on the QoL of caregivers in heart transplant patients who have a unique treatment and care process.

Response: I did not discuss the effects of specific conditions such as immunosuppression, rejection, infection, and biopsy procedures requiring frequent hospitalization on the QoL of caregivers in heart transplant patients who have a unique treatment and care process since in the present sample there were no heart transplanted patients. But if you believe it is important then I will do it with pleasure

Reviewer 2 Report

Comments and Suggestions for Authors

This article addresses an important clinical issue by examining the factors that affect the quality of life (QoL) of caregivers of patients with heart failure, from both the caregiver’s and patient’s perspectives. QoL was assessed using the SF-36 survey in a relatively large sample (n=340), and the statistical analysis is detailed. The overall structure is solid, and the discussion is consistent with the literature and includes practical clinical recommendations. However, the following revisions are recommended:

1- If possible, include SF-36 subscale scores either in the main text or as a supplementary table to provide a more granular understanding of which dimensions of QoL are most impacted.

2- Simplify the discussion section by removing repetitive statements and focusing on the most critical findings.

3-Emphasize effect sizes more clearly (e.g., explain the clinical significance of a 4.6-point increase in mental component score).

4-Expand the limitations section to better address methodological concerns, particularly the use of convenience sampling and the potential for social desirability bias.

5-A thorough language edit is strongly recommended, ideally by a native English speaker, to improve clarity and professionalism throughout the manuscript.

Comments on the Quality of English Language

The English in the manuscript is mostly understandable, but there are many small grammar and wording mistakes that make some parts unclear. A professional language edit is recommended. Here are a few examples:

-Instead of “those who were a lot insecure”, it should be “those who felt very insecure.”

-Instead of “those who little informed”, use “those who were poorly informed.”

-The sentence “caregivers remain in hospital with patient had worse QoL” could be clearer as “caregivers who stayed in the hospital with the patient had worse QoL.”

Author Response

Reviewer 2

Thank you very much for your time ! I did all you asked me. I write here all modifications which are better presented within the text in highlight.

  1. If possible, include SF-36 subscale scores either in the main text or as a supplementary table to provide a more granular understanding of which dimensions of QoL are most impacted.

ANSWER : Higher scores indicate better health status, and  a mean score of 50 is articulated as a normative value for all scales. Both summaries are impacted but most impacted is physical component where the score is 33.1±6.9

  1. Simplify the discussion section by removing repetitive statements and focusing on the most critical findings.

ANSWER : Done

3-Emphasize effect sizes more clearly (e.g., explain the clinical significance of a 4.6-point increase in mental component score).

ANSWER : Multiple regression showed that caregivers with university education had 4.6 points better mental QoL score, which is a finding of clinical significance. Educated individuals’ cultivate skills and attitudes that enable them to better comprehend their needs and make informed decisions. Education per se is a protective determinant of mental health. A year of education lead to a lower likelihood of depression and anxiety, through several pathways such as improved health behavior and knowledge [29].

4-Expand the limitations section to better address methodological concerns, particularly the use of convenience sampling and the potential for social desirability bias.

ANSWER : Convenience sampling involves collecting samples from easily accessible individuals and not selected through a randomized process where every individual has equal chance of inclusion in the sample. Convenience sampling may not reflect the characteristics of the entire caregivers-population

5-A thorough language edit is strongly recommended, ideally by a native English speaker, to improve clarity and professionalism throughout the manuscript.

Comments on the Quality of English Language

The English in the manuscript is mostly understandable, but there are many small grammar and wording mistakes that make some parts unclear (I Modified). A professional language edit is recommended. Here are a few examples:

-Instead of “those who were a lot insecure”, it should be “those who felt very insecure.”

ANSWER :  those who were a lot insecure”. The word “a lot” belongs to Likert scale

-Instead of “those who little informed”, use “those who were poorly informed.”

ANSWER The word “little” belongs to Likert scale

-The sentence “caregivers remain in hospital with patient had worse QoL” could be clearer as “caregivers who stayed in the hospital with the patient had worse QoL.”

ANSWER: We have updated to Worse QoL only in mental component had caregivers who stayed in hospital with the patient

Round 2

Reviewer 1 Report

Comments and Suggestions for Authors

I have carefully read the revised version of the article titled "Factors affecting quality of life of caregivers of patients with heart failure" that was sent to me for re-examination. First of all, I would like to congratulate the authors for the effort they have put into re-editing this article based on the comments and suggestions of the reviewers. I can say that with this new version, the article has become more comprehensive and contributes to the literature.

Best regards